# Assessing Annoyance and Sleep Disturbance Related to Changing Aircraft Noise Context: Evidence from Tan Son Nhat Airport

**DOI:** 10.3390/ijerph22081296

**Published:** 2025-08-19

**Authors:** Thulan Nguyen, Tran Thi Hong Nhung Nguyen, Makoto Morinaga, Yasuhiro Hiraguri, Takashi Morihara

**Affiliations:** 1Department of Architecture, Faculty of Engineering, Osaka Metropolitan University, 3-3-138 Sugimoto, Sumiyoshi-ku, Osaka 558-8585, Japan; 2Graduate School of Natural Science and Technology, Shimane University, 1060 Nishikawatsu, Matsue 690-8504, Japan; hongnhungnguyen2109@gmail.com; 3Department of Architecture, Daido University, 10-3 Takiharucho, Minami-ku, Nagoya 457-8530, Japan; morinaga@daido-it.ac.jp; 4Department of Architecture, Kindai University, 3-4-1 Kowakae, Higashiosaka 577-8502, Japan; hiraguri@arch.kindai.ac.jp; 5Department of Architecture, National Institute of Technology, Ishikawa College, Tsubata 929-0392, Japan; morihara@ishikawa-nct.ac.jp

**Keywords:** annoyance, sleep disturbances, aircraft noise, exposure-response relationship, non-acoustic factors, mediator

## Abstract

This study examines the impact of aircraft noise on annoyance and sleep disturbances among residents near Tan Son Nhat Airport in Ho Chi Minh City, Vietnam, from 2019 to 2023. It aims to assess the specific effects of aircraft noise exposure on sleep quality, as well as changes in exposure due to reduced air traffic during the COVID-19 pandemic. Surveys conducted before and during the pandemic revealed that, despite lower noise levels, residents continued to report high levels of annoyance, indicating a complex exposure-response relationship. This study evaluates both the impact of aircraft noise levels and the role of non-acoustic factors in mitigating sleep disturbances and shaping residents’ responses over time. The study’s findings support the applicability of WHO guidelines in this context and emphasize the importance of considering both noise reduction and community engagement in noise management strategies.

## 1. Introduction

Recent studies have increasingly highlighted the growing concern over aircraft noise exposure and its impact on general health and sleep disturbance [1,2,3,4,5,6,7,8,9,10]. Studies show that nighttime aircraft noise disrupts sleep by causing frequent arousals, reducing deep and REM sleep, and leading to daytime fatigue. Long-term exposure is associated with elevated blood pressure, arterial stiffness, and increased risk of cardiovascular disease, partly due to oxidative stress and inflammation. Additionally, residents near airports report high levels of annoyance, which further contributes to stress and poor health. The World Health Organization (WHO) has recommended strict limits on aircraft noise levels to safeguard public health, advising that nighttime aircraft noise (*L*_night_) should be reduced to below 40 dB [11]. A meta-analysis combining 11 new studies *(n* = 109,070) with 25 studies from the original WHO analysis (*n* = 64,090) found that while exposure–response relationships were consistent at lower noise levels, aircraft noise led to significantly greater sleep disturbance at higher levels [12,13]. Interestingly, research from both European and non-European contexts found no significant difference in reported sleep disturbances, indicating a potentially universal biological response to aircraft noise [4,5].

Many developed countries have implemented various measures to limit the number of people affected by aircraft noise, including regulations on aviation operations, airport planning, and expansion strategies related to noise abatement [14]. However, many Asian developing countries have only adopted general environmental noise standards and has yet to establish a legal and regulatory framework specific to aircraft noise. Although data from Vietnam (Hanoi, Ho Chi Minh City, and Da Nang) were referenced in the WHO guidelines, local scatterplots reveal discrepancies with WHO thresholds [15,16,17,18], supporting the interpretation that annoyance levels are influenced not only by noise exposure but also by non-acoustic factors. With annual increases in air traffic noise in Asia, it is essential to develop policy frameworks that address both the acoustic and non-acoustic contributors to health impacts. This study aims to assess the dynamic relationship between aircraft noise pollution and urban residents’ health, emphasizing the need for sustainable noise management strategies to mitigate potential adverse effects.

Tan Son Nhat International Airport (TSN), located in a densely populated urban area, is the busiest airport in Vietnam and has some of the highest levels of noise exposure. Unlike many airports in developed countries that enforce nighttime curfews to protect residents’ sleep, Tan Son Nhat International Airport (TSN) operates flights around the clock. Due to high demand for passenger and cargo transport, and in the absence of curfew regulations, flights occur regularly throughout the night. This continuous operation results in significant nighttime aircraft noise exposure for surrounding neighborhoods, making sleep disturbance a critical health concern in the residential areas around TSN.

During the COVID-19 pandemic in 2020, travel restrictions significantly reduced air traffic, resulting in a notable drop in noise levels. This presented a unique opportunity to study the effects of noise intervention. This study investigates the changes in aircraft noise exposure caused by fluctuations in flight operations between 2019 and 2023, focusing on their impact on annoyance and sleep disturbance among residents living near TSN. The COVID-19 pandemic in 2020 led to a sharp decline in flight activity, resulting in a temporary reduction in noise exposure—an effect documented through surveys conducted in June and September 2020. By 2023, air traffic had largely returned to pre-pandemic levels, prompting a follow-up health survey in August 2023 to assess the long-term effects of these changes in noise exposure. This study aims to address the following research questions:

How did residents react to the fluctuations in aircraft noise exposure caused by the reduction in air traffic during the COVID-19 pandemic and its subsequent recovery?

How do non-acoustic factors influence the relationship between aircraft noise exposure and residents’ responses, particularly in terms of annoyance and sleep disturbances?

How relevant and applicable are the WHO noise exposure guidelines in the context of a developing country like Vietnam, where both exposure levels and community characteristics may differ from those in more developed nations?

By analyzing both acoustic and non-acoustic factors, this study aims to better understand how these variables interact to influence sleep disturbance and overall health. The findings are expected to contribute to the development of more effective and context-sensitive noise management strategies, as well as to inform discussions on the applicability of international noise guidelines in developing country contexts.

## 2. Materials and Methods

### 2.1. Survey Plan and Areas

The first survey at TSN was done in 2019 (1st survey), followed by surveys 3 months (2nd survey), 6 months (3rd survey), and 3 years (4th survey) after the onset of the COVID-19 pandemic in early 2020. The 2nd and 3rd surveys examined health impacts due to reduced noise around the airport compared to 2019. The 4th survey investigate the trend of response change when flight operations had largely recovered in August 2023 to assess the long-term effects of noise exposure.

TSN has two parallel runways (07L–25R and 07R–25L) running in the east-west direction. During all survey periods, aircraft consistently used the same takeoff and landing directions, with landings occurring from the east and takeoffs toward the west. The 12 sites, S1 to S12, have been selected for investigation (Figure 1). During all survey periods, aircraft consistently used the same takeoff and landing directions, with landings occurring from the east and takeoffs toward the west. Among them, sites S1 to S5 are located directly under the aircraft landing path, while sites S6 to S9 are under the aircraft takeoff path. Site S10 is situated south of the runway, and sites S11 and S12 are designated as control areas because these areas experience minimal impact from aircraft noise. Locations along the runway were selected at various distances to obtain a wide range of aircraft noise exposure levels. The surveyed residential areas were situated in the backyards of a transport road. Site inspections confirmed that aircraft noise was the dominant environmental noise source and other potential sources, such as road traffic or construction, were either not present or remained minimal across sites.

### 2.2. Questionnaire Survey

The survey was conducted face-to-face on weekends with one adult (aged 18 or older) per household, selected based on availability. Interviewers visited all residences in the selected areas to collect responses. All interviewers underwent standardized training prior to data collection to ensure consistency and authenticity in administering the questionnaire. The training included detailed familiarization with the survey content, adherence to the ISO/TS 15666 protocol, and specific instructions on maintaining neutrality and avoiding interviewer bias. A scripted protocol was used to ensure that all questions were presented uniformly across participants. At the start of the interview, participants were informed about the study objectives and assured of the confidentiality and voluntary nature of their participation.

The questionnaire was designed in accordance with ISO/TS 15666 [19] to examine the effects of aircraft noise exposure alongside other potential influencing factors. The key outcomes assessed were:(i)Annoyance: Measured by the percentage of respondents who were highly annoyed (%HA), defined as those selecting a rating of 8, 9, or 10 on an 11-point numerical scale.(ii)Insomnia: Assessed using the Insomnia Symptom Questionnaire (ISQ) [20,21]. Respondents were identified as experiencing insomnia (%ISM) if they reported difficulty sleeping or excessive daytime sleepiness more than three times per week, along with at least one additional insomnia symptom with the same frequency.

The questions and scales used to assess annoyance and insomnia in the surveys are summarized in Table 1. In addition, the questionnaire collected data on self-reported health status to explore the broader impacts of aircraft noise exposure. The results from all surveys were compared to evaluate temporal trends in annoyance, sleep disturbances, and insomnia associated with aircraft noise exposure around TSN.

### 2.3. Exposure Levels

Noise exposure levels for all surveys were estimated using the Integrated Noise Model (INM) version 7.0, developed by the U.S. Federal Aviation Administration (FAA). INM simulates aircraft noise based on flight operations data, aircraft type, operational profiles, and environmental conditions. The prevailing westerly wind conditions at TSN result in the dominant use of westward flight paths for both arrivals and departures. The estimation was based on flight data logged over a 7-day survey period, compared with seasonal average traffic to ensure typical noise conditions. For each of the four survey periods (2019, June 2020, September 2020, and August 2023), a 7-day representative flight log was collected to model typical noise conditions. These logs included aircraft identification, types, takeoff/landing times, runway used, and actual flight route data captured via an ADS-B (Automatic Dependent Surveillance–Broadcast) system at the airport office, providing real-time positional data at one-second intervals. Noise contours were generated by inputting this flight data into the INM model, using aircraft performance profiles and standard noise emission characteristics from the INM database. Population and residential building geolocation data (mapped using GIS) were overlaid on the noise contours, allowing assignment of exposure levels (in dB) to each surveyed household. Day-evening-night-weighted sound pressure level (*L*_den_) and Night-time equivalent continuous sound pressure level (*L*_night_) were calculated based on aircraft types and operations during each period. The resulting contour maps reflect spatial variation in noise exposure around TSN Airport across the four survey periods and form the basis for exposure classification in the statistical analysis.

In the 2019 surveys, on-site measurements were conducted to assess environmental noise exposure and to validate noise prediction models. These measurements focused on estimating *L*_den_ and *L*_night_ at representative locations around TSN. A total of twelve measurement points were selected, covering a range of distances and directions from the airport to capture variation in aircraft noise exposure. At each point, sound levels were continuously measured for seven consecutive days using class 2 sound level meters (NL-42, RION, Tokyo, Japan), set to log sound pressure levels (*L*_p_) at 100 miliseconds intervals. Microphones were mounted on rooftops at a height of 1.5 m above the roof surface and at least 1 m away from any reflective surfaces, in accordance with ISO 1996-2:2017 guidelines [22]. Each microphone was fitted with an all-weather windscreen to minimize wind-induced noise.

During the measurement periods, meteorological conditions were confirmed from the data of local weather station. Measurements were only conducted under favorable conditions, avoiding heavy rain or strong wind. Wind speeds remained below 5 m/s, and the prevailing wind direction was from the west, aligning with typical flight operations at TSN Airport. Simultaneously, detailed aircraft operation data were collected using ADS-B systems set up in the airport office, capturing real-time flight paths, aircraft types, altitudes, and timestamps. These data were used to associate specific noise events with aircraft activities. *L*_den_ and *L*_night_ were calculated from the measured *L*_p_ data following the European Environmental Noise Directive (END) methodology. Specifically, *L*_den_ was computed by applying penalty weights of +5 dB for evening hours (19:00–22:00) and +10 dB for night-time hours (22:00–06:00), while *L*_night_ was calculated as the equivalent continuous sound level over the full night-time period. The measured values were then compared with the predicted noise exposure levels generated by INM to assess the accuracy of the model. Subsequent exposure assessments in this study are based on the validated predicted *L*_den_ and *L*_night_ levels.

## 3. Results

### 3.1. Demographic Data of Respondents

Table 2 summarizes the demographic data of respondents from all surveys conducted around TSN, with response rates ranging from 28.9% to 70.8%. The June 2020 survey had the lowest response rate at 28.9%, primarily due to external factors such as the COVID-19 pandemic, which disrupted daily life and limited people’s ability to participate. Despite this fluctuation, response rates remained relatively high across the other surveys. In all surveys, the proportion of women was slightly higher than that of men, a trend that remained consistent across the years. Additionally, respondents aged over 60 years accounted for less than 30% of the total number of respondents in all surveys, reflecting Vietnam’s generally younger population. There were no significant differences in demographic data between the ‘before’ and ‘after’ surveys, suggesting that the sample groups remained relatively stable despite the challenges posed by the pandemic. These demographic trends align with the broader characteristics of Vietnam’s population, including a youthful demographic and a slightly higher proportion of women [23].

### 3.2. Changes in Flight Operations and Noise Levels

Table 3 summarizes the average number of daily flights during the four survey periods. The number of flight operations at TSN significantly changed throughout the survey period. The number of flights at TSN was 728 flights per day in 2019, led to very high levels of aircraft noise exposure around the airport, directly affecting the surrounding communities. In March 2020, the government ordered the suspension of international flights due to the COVID-19 pandemic. This decision led to a drastic reduction in flight numbers at TSN. During the second survey period (June 2020), the daily flight count dropped to 413 and further declined to 299 by the third survey in September 2020. This decrease in flight numbers was in line with the resurgence of COVID-19 cases in July 2020, which resulted in more domestic flight restrictions. By 2023, flight numbers had nearly returned to pre-pandemic levels, approaching the 728 flights per day seen in 2019. Table 4 summarizes estimated noise levels for all surveys with the noise exposure measured in 2019 to confirm the validity of the estimation. Fluctuations in the number of flights at Tan Son Nhat Airport directly influenced the levels of aircraft noise exposure in surrounding areas, with significant reductions during the pandemic followed by a gradual recovery. Figure 2 and Figure 3 depict the projected areas exceeding 55 dB *L*_den_ and 40 dB *L*_night_ respectively, and reflect the changes in airport-related noise exposure during and after the COVID-19 pandemic. The 55 dB *L*_den_ threshold and 40 dB *L*_night_ threshold were used as reference levels align with the WHO (2018) recommendations for environmental noise and health. In Figure 2, areas with noise levels exceeding 55 dB are wider under the departure path. The variation of noise levels among survey periods was stronger for sites under the departure path than for landing sites. Consistent with changes in flight operations, there was the shrinkage of these contours during the pandemic and their subsequent recovery can be observed in the case of TSN. The nighttime noise map depicts the spatial distribution of areas where nighttime noise levels exceed 40 dB across four survey periods (2019, 2020-June, 2020-September, and 2023-August) (Figure 3). Areas exceeding 40 dB *L*_night_ are primarily concentrated along the departure and arrival paths, but are noticeably wider under the departure routes, reflecting more intense noise emissions from takeoffs compared to landings. During 2020, particularly in June and September, the contours shrank significantly, indicating a sharp reduction in nighttime noise exposure. This corresponds with the drop in nighttime flights (from 133 total in 2019 to just 37 in 2020-September), as shown in Table 3. By August 2023, the nighttime noise contours expanded again, closely resembling the 2019 levels, reflecting the resumption of flight operations (nighttime flights recovered to 145 per day). Populated or residential zones located directly beneath or adjacent to these contours are likely experiencing chronic nighttime noise exposure, which may contribute to sleep disturbance, annoyance, and potential long-term health effects.

### 3.3. The Relationship Between Exposure Levels and Annoyance

Figure 4 illustrate the exposure–response relationships between aircraft noise *L*_den_ and high noise annoyance (%HA) across all four survey periods. For each dataset, a logistic regression model were independently fitted using the probability of a respondent being highly annoyed (%HA) as the dependent binary outcome (defined as respondents rating annoyance 8, 9, or 10 on the ISO/TS 15666 scale) and *L*_den_ as the independent variable. No adjustments were made for non-acoustic or individual factors to maintain comparability with the reference curve proposed in the WHO Environmental Noise Guidelines (2018). The exposure-response (*L*_den_–%HA) relationships of each survey reveals noticeable temporal variation. To evaluate the model fit, the area under the curve (AUC) was calculated for each logistic regression model. The AUC was 0.74382 for the 1st survey in 2019 (pre-pandemic baseline), 0.75062 for the 2nd survey in June 2020 (during early COVID-19 restrictions), 0.43859 for the 3rd survey in September 2020 (later pandemic stage), and 0.77363 for the 4th survey in August 2023 (post-pandemic period). These results indicate that the models had acceptable to good fit in all periods except the 3rd survey, where the notably lower AUC suggests a weakened or inconsistent relationship between noise exposure and annoyance during that time. In the 2nd survey (June 2020), although *L*_den_ were significantly reduced due to flight suspensions during the COVID-19 pandemic, %HA was unexpectedly higher than in 2019, indicating heightened sensitivity possibly linked to anxiety, uncertainty, or increased time spent at home. By the 3rd survey (September 2020), the annoyance curve had dropped significantly, especially within the 50–70 dB range, despite only modest changes in exposure. This suggests a shift in public attitudes or adaptation effects during the later stage of the pandemic.

### 3.4. The Relationship Between Exposure Levels and Sleep Effect

Figure 5 shows the exposure–response relationship between *L*_night_ and insomnia (%ISM). The area under the curve (AUC) for each logistic model was 0.60793 for the 1st survey in 2019, 0.51896 for the 2nd survey in June 2020, 0.53416 for the 3rd survey in September 2020, and 0.68973 for the 4th survey in August 2023. These values suggest a generally weak model fit, particularly during the pandemic period. Patterns here mirror those in the %HA data but with notable differences. The %ISM curves remain relatively flat from the 1st to 3rd surveys, suggesting that sleep disturbance was less sensitive to changes in nighttime noise or that residents were resilient during the pandemic. However, the 4th survey (August 2023) shows a sharp increase in %ISM across all *L*_night_ levels. This suggests that sleep sensitivity increased substantially in the post-pandemic context, potentially due to disrupted circadian routines, a return to normal working hours, or heightened stress.

The exposure–response relationships obtained in this study were compared with those established with the WHO 2018 guideline curve for %HSD. It is important to note that while the WHO 2018 guideline provides a curve for %HSD in relation to *L*_night_, our survey uses %ISM based on self-reported insomnia symptoms. Thus, comparisons with the WHO curve are indicative rather than directly equivalent. Figure 5 shows that the 4th survey (August 2023) exhibited a sharp rise in %ISM with increasing *L*_night_, approaching or even exceeding the WHO 2018 %HSD curve at higher exposure levels. This suggests that post-pandemic lifestyle changes may have increased sleep sensitivity. In contrast, surveys during 2020 (2nd and 3rd) show a relatively flat response curve, potentially due to altered sleep patterns and reduced exposure to cumulative noise sources during COVID-19 restrictions.

In all four surveys conducted around TSN, both %HA and %ISM levels remained consistently lower than the WHO reference curves, despite temporal variations. This finding aligns with previous studies conducted under stable noise conditions [5], but in the case of TSN, the patterns can be attributed to distinct environmental and contextual factors. TSN is centrally located in a densely populated area of Ho Chi Minh City, where residents are routinely exposed to high levels of urban noise, particularly from road traffic. This situation appears to shape residents’ perceptions of aircraft noise, potentially increasing tolerance and resulting in lower reported annoyance and sleep disturbance levels.

### 3.5. The Impact of Non-Acoustic Factors

#### 3.5.1. Multiple Logistic Regression Models

To further explore the patterns of exposure-response relationships, the TSN survey model incorporated non-acoustic factors to examine how residents’ reactions to noise changed over time and explain why lower annoyance and insomnia responses were observed compared to WHO 2018 reference curvess. Multiple logistic regression models were constructed to examine the effects of non-acoustic factors. Data on non-acoustic factors collected in the surveys include residential environment, individual characteristics, and attitudes, all believed to influence reactions to aircraft noise. All statistical analyses were conducted using JMP 13.0. It should be noted that the data from the second survey conducted in June 2020 were not included in the analysis, as the low response rate produced an insufficient and statistically unreliable dataset compared to the other survey waves.

Initially, analyses were conducted by incorporating independent variables comprising both noise levels and non-acoustic factors. The goal was to create models that offer the best overall measure of fit for data obtained at TSN, including variables significantly related to the likelihood of annoyance and insomnia. Consequently, separate models were developed for the TSN surveys, with different variables used to construct each regression model (Table 5 and Table 6). Only non-acoustic factors that showed significant associations with annoyance and insomnia were retained in the final models, while basic demographic attributes such as age and gender were excluded due to lack of significance.

In this analysis, the dependent variable was binary (1 = “highly annoyed”/0 = “not highly annoyed” or 1 = “have insomnia symptom”/0 = have no insomnia symptom”). The independent variables included a continuous variable for noise exposure (*L*_den_/*L*_night_) and other nominal (dummy-coded) non-acoustic variables. To examine changes in annoyance and insomnia over time, the survey year was included as a nominal variable. Specifically, the outcomes from the 3rd and 4th surveys were compared against the 1st survey, which served as the reference category. This approach allowed the analysis to identify whether significant differences in annoyance or insomnia existed in later surveys relative to the baseline (1st survey), without assuming a linear trend across all survey years.

(i)Annoyance model:

As shown in Table 5, non-acoustic factors such as short residence duration (*p* ≤ 0.05), poor views (*p* ≤ 0.001), and small floor area (*p* ≤ 0.001) were associated with higher prevalence of annoyance. These findings indicate that residential conditions and visual environment play an important role in shaping noise annoyance responses. Regarding survey factors, the 3rd survey (2019) showed a marginally significant increase in the likelihood of being highly annoyed compared to the 1st survey (*p* ≤ 0.05), while the 4th survey (2023) did not show a statistically significant difference. A key finding was the interaction between noise exposure and survey factors (*L*_den_ × Survey). The interaction term between *L*_den_ and the 3rd survey was statistically significant (*p* ≤ 0.05), and the interaction with the 4th survey was even more significant (*p* ≤ 0.01). These results indicate that the effect of survey year on annoyance varies with the noise level: as noise exposure decreases, the effect of the survey factor increases, and vice versa. Notably, the negative coefficient for the interaction with the 3rd survey suggests that the influence of survey-related factors was stronger at lower noise levels. These findings point to complex, time-dependent factors influencing annoyance, and suggest that even in contexts of reduced noise exposure (e.g., in 2020 and 2023), other environmental or social variables may sustain or elevate annoyance levels.

(ii)Insomnia model:

The insomnia (ISM) model showed that night-time noise exposure (*L*_night_) had a significant effect on the prevalence of insomnia (*p* ≤ 0.05), indicating that even modest increases in night-time noise were associated with a higher likelihood of reporting insomnia symptoms. While the survey factor for the 3rd survey (2019) was not statistically significant, the 4th survey (2023) showed a significant increase in the prevalence of insomnia compared to the 1st survey (*p* ≤ 0.01), suggesting that contextual factors following the pandemic may influence sleep outcomes. However, the interaction between noise exposure and survey factors (*L*_night_ × Survey) were not significant, indicating that the influence of night-time noise on insomnia did not vary significantly across survey years. Among the non-acoustic variables, several showed significant associations with insomnia. Participants who reported being sensitive to heat were significantly more likely to report insomnia (*p* ≤ 0.05), and those who spent more than 15 h per day at home were also at greater risk (*p* ≤ 0.05). In addition, respondents with a bad view from home were significantly more likely to report insomnia symptoms (*p* ≤ 0.01). These findings highlight that non-acoustic environmental and lifestyle factors—such as increased time at home due to pandemic-related restrictions and thermal sensitivity—may contribute to sleep disturbances, even in the context of an objectively quieter acoustic environment.

#### 3.5.2. Structural Equation Model

The structural equation modeling (SEM) method was applied to understand the complex relationship between social responses to noise and various factors. The SEM model was constructed by integrating questionnaire items from all surveys conducted at TSN. Initially, separate SEM models were constructed for the 2019, 2020, and 2023 datasets to examine how the relationships among variables evolved over time. Based on these individual models, a process of refinement was undertaken to identify the most influential and consistent factors across years. This led to the development of a unified model that captures the structural relationships between noise exposure, mediating factors, and health-related outcomes. Table 7 presents the questions and evaluation scales used for measuring moderating variables in the model.

Figure 6 illustrates the final structural equation model (SEM) that fit all the surveys. In this model, latent variables (represented by ellipses) refer to parameters that are not directly observed but inferred from measurable indicators, while observed variables (represented by rectangles) are derived directly from survey responses. In the SEM model, noise level (*L*_den_ and *L*_night_) was included as an exogenous exposure variable. Mediating factors, such as living conditions and health status, were modeled to reflect the pathways through which noise exposure may affect outcomes like annoyance and insomnia. Key components of the model include:

(1)Noise Exposure *L*_den_ and *L*_night_: Used as the primary exposure metric, reflecting day-evening-night noise levels and nighttime noise levels.(2)Living Conditions: A latent construct inferred from residents’ evaluations of their view, access to green space and street sceneries, from their house. These environmental quality indicators help explain how surroundings influence noise perception and tolerance.(3)Health: This latent factor is constructed from responses related to Stress, self-rated health, and exercises concern, providing an overall picture of residents’ well-being.(4)Sensitivity: Comprising noise sensitivity, vibration sensitivity, and odor sensitivity, this factor captures individual predispositions to environmental stimuli, which may amplify the effects of noise exposure.(5)Length of time at home: An observed variable based on the number of hours residents spend at home. This behavior was influenced by lockdowns during the pandemic and affects the level of noise exposure, thereby having direct implications for sleep and annoyance outcomes.(6)Outcomes: The two primary reported outcomes—annoyance and insomnia—are observed factors reflecting the social and health responses to aircraft noise.

**Figure 6 ijerph-22-01296-f006:**
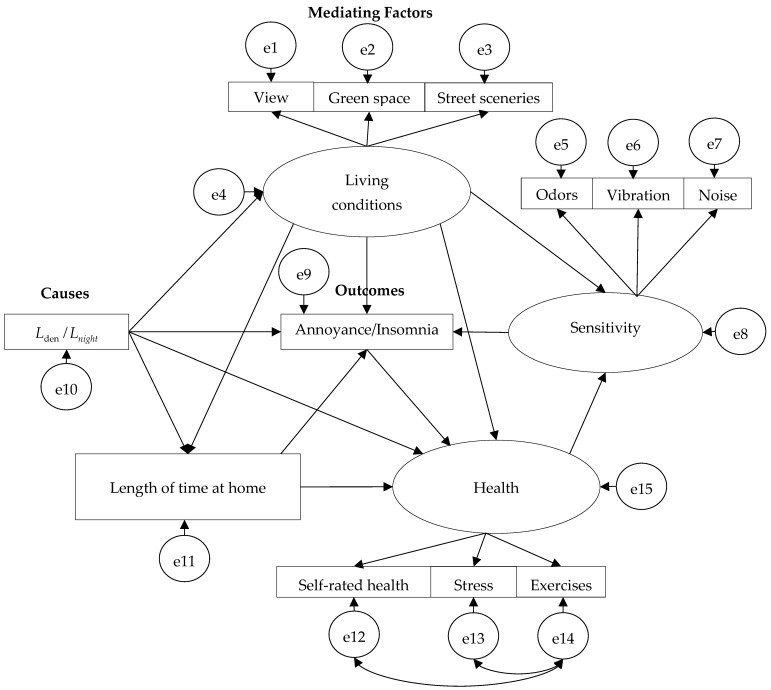
The structural equation model (SEM) developed by integrating the questionnaire items from all the survey.

To obtain the annoyance model, we followed a stepwise process. First, questionnaire data on annoyance and related variables from each survey year (2019, 2020, and 2023) were merged with corresponding aircraft noise exposure metrics *L*_den_. Separate SEMs were developed for each survey year to capture year-specific relationships between noise exposure, mediating factors, and reported annoyance. From these individual models, we identified consistent and statistically significant pathways across years, refining the variable set to include only robust predictors. These variables, both latent and observed, were then integrated into a unified SEM that reflects the most stable structural relationships over time. This unified model was subsequently evaluated for model fit and predictive validity using established SEM fit indices.

The structural equation modeling results (Figure 7 and Table 8) depicts how the pathways influencing noise annoyance evolved over three phases: before the COVID-19 pandemic (2019), during the pandemic (2020), and in the recovery phase (2023). Key statistical indicators show good model fit (χ^2^ = 446.203, *p* < 0.01, df = 126, GFI = 0.932, CFI = 0.906, and RMSEA = 0.052), confirming the model’s reliability across all three years.

In 2019 Model (Pre-Pandemic), direct effect of noise (*L*_den_) on annoyance was statistically significant (Estimate = 0.105, *p* < 0.001). Living conditions influenced sensitivity (*p* < 0.001), which in turn influenced annoyance (*p* < 0.001). The direct effects of noise, sensitivity, living conditions, and annoyance on health were significant. Under normal circumstances, annoyance was largely determined by the physical noise levels and sensitivity.

In 2020 Model (Post-Pandemic), the direct effect of *L*_den_ on annoyance was statistically significant (*p* = 0.031). Length of time at home emerged as a significant predictor of annoyance (Estimate = 1.003, *p* < 0.001), reflecting lockdown conditions and increased home confinement. Living conditions and sensitivity had no significant influence, possibly because unusual pandemic constraints overrode contextual evaluations. During COVID-19, annoyance became less influenced by environmental or acoustic factors and more shaped by personal lifestyle disruptions and increased noise exposure due to time spent at home.

In 2023 Model (Post-pandemic Recovery), the direct path from *L*_den_ to annoyance re-emerged as significant (Estimate = 0.122, *p* < 0.001), indicating a partial return to pre-pandemic perception patterns. Length of time at home also remained significant (Estimate = 0.342, *p* = 0.012), suggesting lingering effects of changed living routines. Health was significantly effected by sensitivity, annoyance, and length of time at home, though the direct path from *L*_den_ to remained non-significant.

The annoyance model suggests that under normal circumstances, residents’ reactions to aircraft noise were more strongly shaped by their perceptions of the surrounding environment and their personal susceptibility to environmental stimuli. During the pandemic, health concerns and length of time spent at home became significant factors influencing annoyance. The COVID-19 outbreak led to lifestyle changes, including increased time spent indoors and heightened health awareness. In this context, annoyance became less associated with general environmental conditions and more influenced by health-related stressors and domestic exposure. The annoyance response in the recovery phase reflects a hybrid pattern: objective noise exposure has regained influence, but pandemic-induced behavioral changes still contribute, particularly through increased noise exposure at home.

Insomnia model:

The structural equation modeling (SEM) in Figure 8 reveal how factors contributing to insomnia changed before, during, and after the COVID-19 pandemic. Key statistical indicators show good model fit (χ^2^ = 348.733, *p* < 0.01, df = 126, GFI = 0.945, CFI = 0.930, and RMSEA = 0.043), confirming the model’s reliability across all three years. Table 9 shows parameter estimates of the model for insomnia.

In 2019 model (Pre-Pandemic) nighttime noise (*L*_night_) had no direct effect on insomnia (*p* = 0.216), though it marginally affected health (*p* = 0.065). Health was weakly linked to sensitivity (*p* = 0.046), but its overall role in insomnia was minor. In this phase, insomnia was more influenced by subjective factors like length of time at home. living conditions and individual noise sensitivity rather than by objective noise exposure itself.

In 2020 model (During Pandemic), The path from *L*_night_ to Insomnia became significant and positive (Estimate = 0.010, *p* < 0.001). Time at home and Insomnia relation was non-significant, as was sensitivity, possibly due to pandemic-related psychological overload. Path between *L*_night_ and Time at home was strongly significant (0.016, *p* < 0.001), showing how noise exposure increased as people stayed home more. All the other paths to insomnia were non-significant. The shift suggests that nighttime noise exposure directly disrupted sleep during lockdown, when people spent more time at home and had fewer coping mechanisms. The usual buffer factors (sensitivity, living conditions) lost influence, possibly overshadowed by the extraordinary context of the pandemic.

In 2023 model (Post-Pandemic Recovery), Insomnia was now influenced by multiple significant predictors *L*_night_ (*p* = 0.045), Time at home (*p* < 0.001), and Sensitivity (*p* < 0.001) Health linked to Insomniasiginificantly. The living conditions and Insomnia path was no longer significant, unlike in 2019. In 2023, the model reflects a complex interaction: noise exposure (*L*_night_) directly impacts insomnia, but sensitivity and increased time at home amplify the effect. The role of health defined here as sleep disturbances, stress, and nutritional concerns is more pronounced, acting as a mediator between sensitivity and sleep quality.

Overall, despite reduced aircraft noise during 2020, insomnia levels did not significantly improve, suggesting non-acoustic stressors (e.g., health concerns, stress, home confinement) had strong influence. Post-pandemic, insomnia remains influenced by both acoustic (*L*_night_) and non-acoustic factors, reflecting enduring changes in lifestyle and sensitivity. This supports the hypothesis that personal health and psychological sensitivity amplify the impact of environmental noise on sleep, especially under prolonged exposure or stress.

It is worthy noted that health variables might amplify the negative effects of noise on both annoyance and insomnia. This helps explain why the percentage of people experiencing high annoyance or insomnia did not significantly decrease despite a major reduction in aircraft noise. The persistence of these symptoms despite lower noise levels suggests that non-acoustic factors, particularly personal health, played a critical role in shaping residents’ responses to noise in post-pandemic phase, particularly in terms of their sensitivity to noise and overall sleep quality.

## 4. Discussion

This study confirms that annoyance and insomnia linked to aircraft noise are influenced by both acoustic and non-acoustic factors, echoing trends seen internationally. While noise exposure levels (*L*_den_) decreased significantly during the COVID-19 pandemic due to fewer flights, annoyance levels among residents increased. Similarly, although *L*_night_ levels declined, %ISM did not significantly drop in 2020 and instead increased in 2023,This aligns with a study by Tong et al. (2021), which reported a rise in noise complaints during lockdowns, despite objectively quieter environments [24]. Although actual noise exposure from traffic decreased, complaints about construction and neighborhood noise rose substantially, particularly in areas with higher unemployment and lower housing prices. Despite reduced aircraft operations and lower *L*_den_ values in the 2nd survey (June 2020), the proportion of highly annoyed residents (%HA) remained elevated compared to the pre-pandemic period. This supports our observation that contextual variables, such as housing quality, stress, and increased time spent indoors, can elevate noise sensitivity even when acoustic levels decrease. Similarly, Mitchell et al. (2021) found that lockdown-induced changes in urban soundscapes led to less eventful and more pleasant perceptions in areas that were previously traffic-dominated [25]. This shift, however, did not occur in areas with dominant natural or human sounds. In Aletta et al. (2020), acoustic measurements in 11 London locations revealed a mean 5.4 dB drop in noise level during lockdown [26]. However, the reduction varied significantly by site, indicating the role of urban typology and existing soundscapes in shaping perceived change. This case study in Greater London suggested that this increase stemmed from heightened expectations for quietness and increased occupancy at home, factors also applicable in our TSN context. While actual aircraft noise in TSN reduced in 2020, the impact on residents’ annoyance varied by factors like building insulation, indoor space availability, and noise sensitivity. In the TSN context, where nighttime aircraft operations resumed without curfews, the increase in insomnia by 2023 likely reflects a delayed re-sensitization after a period of reduced exposure, consistent with patterns of shifting perception found in that study. Our findings also support the observations of Guski et al. (2017), who emphasized that psychological and contextual variables, such as individual noise sensitivity, perceived control, and building characteristics, play a critical role in perceived annoyance [12].

In contrast to annoyance trends, insomnia complaints did not rise significantly during the pandemic but showed a noticeable increase in 2023, after flight activity resumed. This pattern differs slightly from Basner et al. (2011), who reported immediate sleep disruption effects with increased aircraft noise [27]. However, our results align with more recent longitudinal studies (e.g., Clark et al., 2020), which suggest that the health effects of noise on sleep can manifest gradually and be amplified after periods of reduced exposure, potentially due to changes in baseline tolerance or expectation [28].

With temporal analysis of exposure–response relationships, this study captures how annoyance and sleep disturbance evolve differently in response to both objective noise exposure and changing social or environmental contexts. This perspective is particularly relevant in developing urban environments, where acoustic environments shift rapidly and residents’ expectations are not fixed. Additionally, by combining multi-year survey data with structural equation modeling, this study provides new empirical evidence from Southeast Asia, a region underrepresented in the aircraft noise literature. Our SEM also affirmed previous findings indicating that noise sensitivity and indoor environment quality mediate the relationship between exposure and health outcomes [29,30]. When assessing environmental noise impacts, it is critical to consider a broader set of interrelated factors including acoustic, psychological, and environmental ones.

Adding to this, the public opinionon noise suring lock down in Turkey confirmed that environmental noise and related annoyance significantly decreased during lockdown, particularly in previously noisy environments [31]. However, annoyance from neighbor noise remained constant, and annoyance related to one’s own dwelling increased, indicating that indoor soundscapes became more salient as people spent more time at home. This reinforces our SEM findings that indoor environment quality was a key mediator in both annoyance and insomnia outcomes in TSN. Particularly during crises like pandemics, the interaction between acoustic exposure and daily living conditions becomes more complex and unpredictable. This emphasizes the need for context-sensitive noise management strategies that extend beyond regulatory thresholds and incorporate qualitative aspects of urban living, especially in rapidly urbanizing cities like Ho Chi Minh City.

## 5. Conclusions

This study examines aircraft noise impacts in a rapidly urbanizing and high-density Southeast Asian city, an underrepresented context in existing literature. By conducting repeated surveys across multiple years during a rare period of drastic flight reduction caused by the COVID-19 pandemic, it captures the temporal dynamics of exposure–response relationships. The study applies a combination of questionaire survey, noise mapping, and structural equation modeling (SEM) to explore how acoustic and non-acoustic factors such as housing, sensitivity, and stress jointly influence annoyance and insomnia. The study revealed clear temporal variations in the exposure–response relationships between aircraft noise and both annoyance (%HA) and insomnia (%ISM) among residents near TSN. Despite overall annoyance levels being lower than those cited in international guidelines such as WHO (2018), annoyance responses fluctuated significantly across the four survey periods. Notably, high noise annoyance increased during the initial phase of the COVID-19 pandemic, even with reduced noise exposure, suggesting that heightened stress, anxiety, and increased time spent at home played key roles. Conversely, sleep disturbance showed a delayed response, with a marked increase in insomnia emerging only in the post-pandemic phase (2023), likely influenced by shifts in circadian routines, a return to regular work hours, and residual stress. Multiple logistic regression and structural equation modeling further underscored the importance of non-acoustic factors—such as poor housing conditions, limited space, stress, and sensitivity—to annoyance and insomnia. By defining the divergence in temporal patterns between annoyance and insomnia, and by contextualizing these outcomes within a post-pandemic urban Southeast Asian setting, this study contributes a nuanced perspective to the international literature. Public responses to noise are not solely determined by decibel levels but are strongly mediated by contextual and personal factors. Therefore, effective noise management and urban planning must incorporate both acoustic and non-acoustic dimensions to better address community health and well-being in rapidly urbanizing environments like Ho Chi Minh City.

## Figures and Tables

**Figure 1 ijerph-22-01296-f001:**
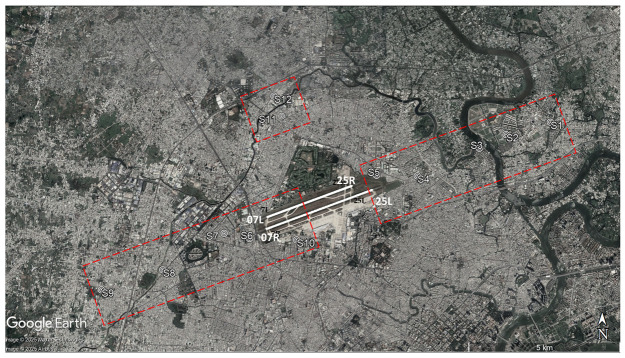
Map of survey sites (Sites 1–12). Locations S1–S12 have been marked on the Google Earth image. Source: © Google Earth.

**Figure 2 ijerph-22-01296-f002:**
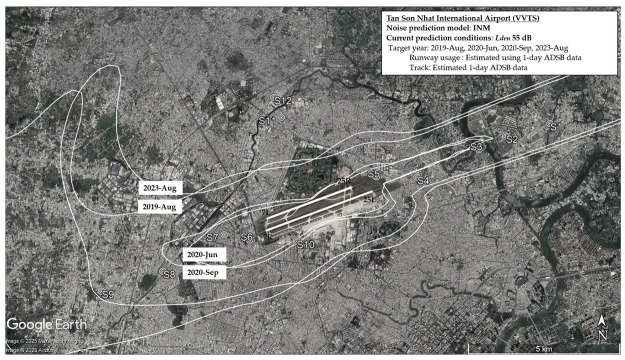
Noise contours of TSN Airport delineate the areas exposed to day-evening-night noise levels exceeding 55 dB across all survey periods.

**Figure 3 ijerph-22-01296-f003:**
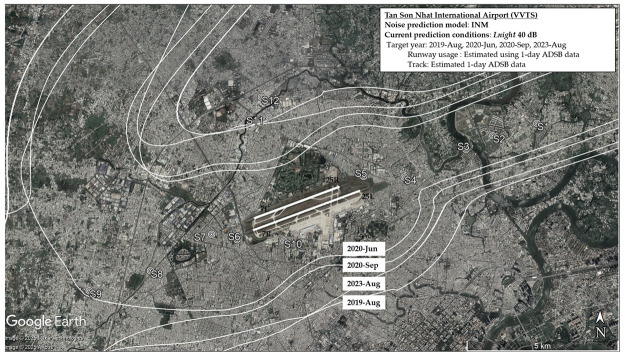
Noise contours of TSN Airport delineate the areas exposed to nighttime noise levels exceeding 40 dB across all survey periods.

**Figure 4 ijerph-22-01296-f004:**
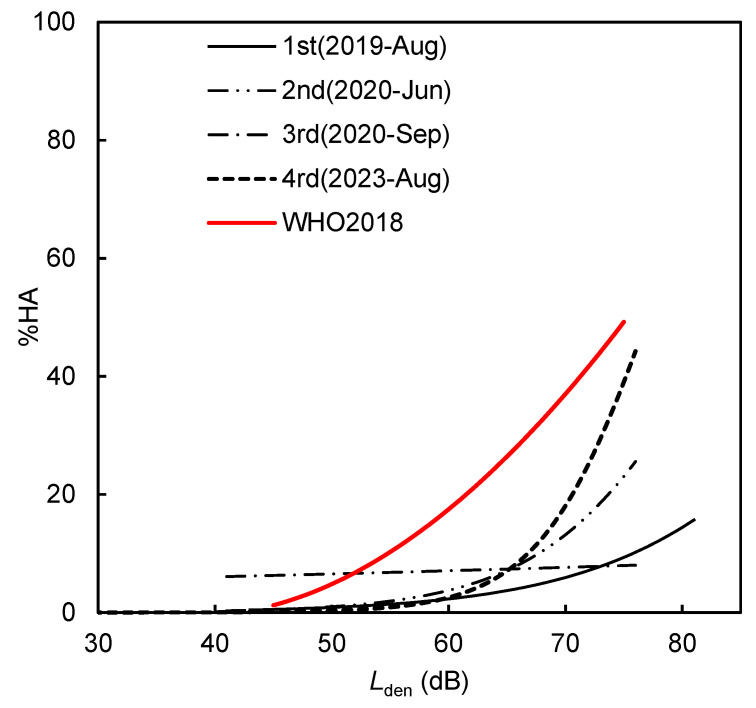
Comparison of *L*_den_–% HA relationships for each survey.

**Figure 5 ijerph-22-01296-f005:**
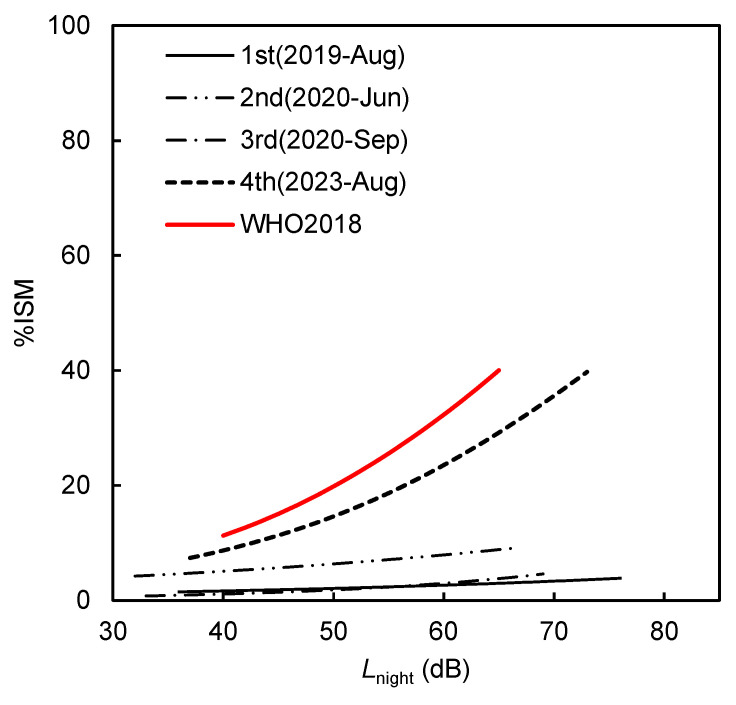
Comparison of *L*_night_–%ISM relationships for each survey.

**Figure 7 ijerph-22-01296-f007:**
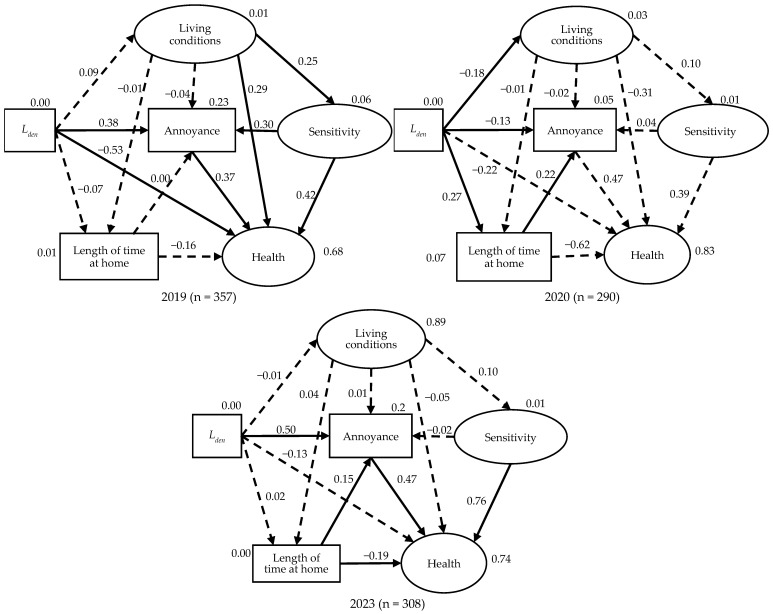
The impact structure in the estimated noise annoyance model in the 2019, 2020 and 2023 surveys using chi-square, GFI, CFI, and RMSEA statistics: chi-square = 446.203, *p* < 0.01, df = 126, GFI = 0.932, CFI = 0.906, and RMSEA = 0.052. Statistically significant paths and standardized regression weights were annotated with (*p* < 0.05). The non-significant paths are represented by dashed lines. The explained variances are annotated for each variable.

**Figure 8 ijerph-22-01296-f008:**
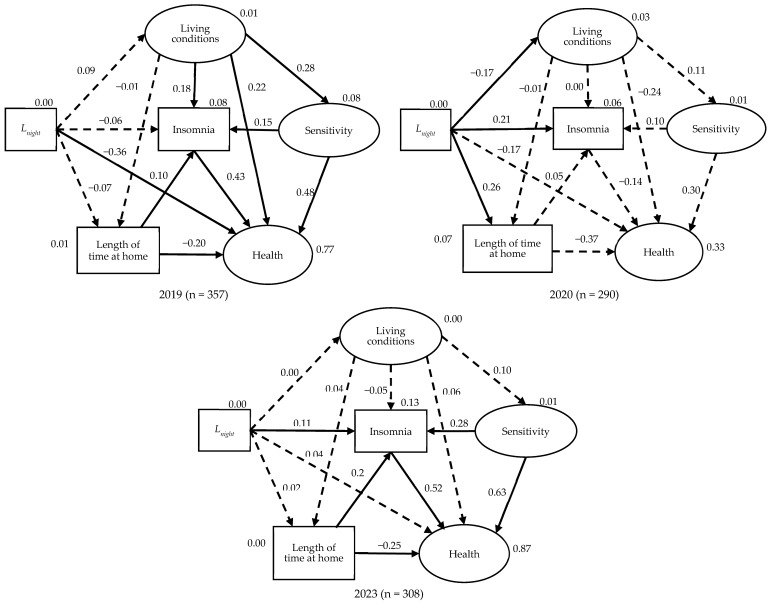
The impact structure in the estimated insomnia model in the 2019, 2020 and 2023 surveys using chi-square, GFI, CFI, and RMSEA statistics: chi-square = 348.733, *p* < 0.01, df = 126, GFI = 0.945, CFI = 0.930, and RMSEA = 0.043. Statistically significant paths and standardized regression weights were annotated with (*p* < 0.05). The non-significant paths are represented by dashed lines. The explained variances are annotated for each variable.

**Table 1 ijerph-22-01296-t001:** The questions and scales used to assess annoyance and insomnia in the surveys.

Measure	Question	Evaluation Scale
Annoyance	Thinking about the last 12 months (1st survey and 4th survey)/three months (2nd survey)/four months (3rd survey), what number from 0 to 10 best shows how much you are bothered, disturbed, or annoyed by aircraft noise?	11-point numerical scale: 0 = not annoyed at all; 10 = extremely annoyed
Insomnia	(a) Do you have any trouble sleeping? No/Yes (b) If “Yes,” please choose the corresponding alternative (Rarely or not at all; Once or twice a week; More than three times a week) for the following items: (1) Difficulty falling asleep (2) Difficulty returning to sleep after awakening during the night (3) Early morning awakening (4) Not feeling refreshed the next morning (5) Daytime sleepiness and inability to work well (6) Others	1 = have no insomnia symptom (*); 2 = have insomnia symptom

Note: (*) Respondents with insomnia symptoms answered “Yes” to (a) “Do you have any trouble with your sleep?”, reported (5) daytime sleepiness and inability to work well more than three times per week, and experienced at least one of the other symptoms (1)–(4) more than three times per week.

**Table 2 ijerph-22-01296-t002:** Demographic Data of Respondents.

		2019	2020 June	2020 September	2023 August	Census (2019) *
Number of respondents		502	145	519	329	
Response rate (%)		60.3	28.9	68.6	70.8	
Sex	Male	46.2	46.5	49.2	48.0	49.9
	Female	53.8	53.5	50.8	52.0	50.1
Age	<60 years old	81.9	70.6	89.9	90.9	88.1
	≥60 years old	18.1	29.4	10.1	9.1	11.9

(*): Adapted with permission from ref. [23] Copyright by General Statistics Office of Vietnam.

**Table 3 ijerph-22-01296-t003:** The average number of daily flight operations.

Time Period		TSN Surveys			
	Operation Models	2019	2020-June	2020-September	2023-August
Day (6:00–18:00)	Arrival	214	140	86	206
	Departure	244	166	121	223
Evening (18:00–22:00)	Total	458	306	207	155
	Arrival	73	45	35	78
	Departure	64	23	20	72
	Total	137	68	55	150
Night (22:00–6:00)	Arrival	77	20	19	76
	Departure	56	19	18	69
	Total	133	39	37	145
All day	Arrival	364	205	140	360
	Departure	364	208	159	365
	Total	728	413	299	725

**Table 4 ijerph-22-01296-t004:** Estimated aircraft noise levels (2019–2023), with measured values shown in parentheses.

Sites	Day-Evening-Night Noise Level (*L*_den_ ^a^)				Night Noise Level (*L*_night_ ^b^)			
	2019	2020-June	2020-September	2023-August	2019	2020-June	2020-September	2023-August
1	64(66)	61	60	63	57(59)	52	52	55
2	65(68)	61	61	64	58(61)	52	53	56
3	66(NA)	60	59	64	58(NA)	51	51	56
4	63(NA)	57	56	61	55(NA)	48	49	54
5	81(79)	76	73	79	74(72)	67	66	72
6	74(74)	71	69	67	66(67)	61	61	59
7	70(70)	65	64	65	62(63)	56	56	57
8	66(NA)	62	62	61	58(NA)	53	54	53
9	64(65)	59	60	56	56(57)	50	52	47
10	67(64)	62	65	68	60(55)	54	57	60
11	47(46)	43	43	47	40(37)	34	36	39
12	45(44)	41	41	45	38(36)	33	34	37

Note: ^a^ Day-evening-night-weighted sound pressure level (*L*_den_) ^b^ Night-time equivalent continuous sound pressure level (*L*_night_); Values in parentheses for 2019 indicate measured data. “NA” = Not Available.

**Table 5 ijerph-22-01296-t005:** The multiple logistic regression of annoyance (Generalized R2: 0.1445; AUC: 0.7596).

Item	Category	Estimate	Std Error	*p*-Value	Odds Ratio	Lower 95%CI	Upper 95%CI
Intercept		−8.023 0	1.495	<0.0001			
*L*_den_ ^a^		0.090	0.022	<0.0001	1.094 ^b^	1.048	1.142
Survey factor	1st survey				1		
	3rd survey	0.419	0.216	*	2.864	1.298	6.317
	4th survey	0.215	0.240	0.3697	2.336	0.982	5.554
*L*_den_ ^a^ × Survey factor	3rd & 1st survey	−0.088	0.028	*			
	4th & 1st survey	0.071	0.033	**			
Duration of residence	>5 years				1		
	≤5 years	−0.357	0.157	*	2.042	1.106	3.773
View from home	Good				1		
	Bad	−0.645	0.175	***	3.633	1.833	7.202
Floor area	>50 m^2^				1		
	≤50 m^2^	0.611	0.160	***	0.295	0.158	0.551

^a^ Day-evening-night-weighted sound pressure level. ^b^ Odds ratio in 1 dB change. *p*-Value ≤ 0.05 = *, *p*-Value ≤ 0.01 = **, *p*-Value ≤ 0.001 = ***.

**Table 6 ijerph-22-01296-t006:** The multiple logistic regression of insomnia (Generalized R2: 0.0875; AUC: 0.7019).

Item	Category	Estimate	Std Error	*p*-Value	Odds Ratio	Lower 95%CI	Upper 95%CI
Intercept		−5.118 0	1.143	<0.0001			
*L*_night_ ^a^		0.046	0.020	*	1.047 ^b^	1.007	1.088
Survey factor	1st survey				1		
	3rd survey	−0.470	0.282	0.0961	0.835	0.318	2.189
	4th survey	0.759	0.249	**	2.855	1.221	6.674
*L*_night_ ^a^ × Survey factor	3rd & 1st survey	0.026	0.030	0.3844			
	4th & 1st survey	0.008	0.027	0.7533			
Heat sensitivity	Not sensitive				1		
	Sensitive	−0.410	0.208	*	2.270	1.004	5.131
Length of time spent at home	Under 15 h				1		
	Over 15 h	−0.388	0.176	*	2.174	1.090	4.337
View from home	Good				1		
	Bad	−0.533	0.191	**	2.901	1.372	6.134

^a^ Night-time equivalent continuous sound pressure level. ^b^ Odds ratio in 1 dB change. *p*-Value ≤ 0.05 = *, *p*-Value ≤ 0.01 = **.

**Table 7 ijerph-22-01296-t007:** Questions and evaluation scales for measuring moderating variables in the model.

Latent Variable	Observed Variable	Question	Scale
	Length of time at home	Thinking about the last four months, how long in a day do you stay at home?	1: Under 8 h 2: 8–15 h 3: Over 15 h hours
Health	Stress	Thinking about the amount of stress in your life, how stressful would you say that most days are?	0: Not at all to 10: Extremely
	Self-rated health	In general, would you say your health is…?	1: Excellent 2: Very good 3: Good 4: Fair 5: Poor
	Exercises	How often do you engage in physical activity over 30 min?	1: Almost everyday 2: 4–5 times a week 3: 2–3 times a week 4: About once a week 5: Once or twice a month 6: Not at all
Sensitivity		In daily life, how sensitive are you to the following climatic factors and environmental conditions	
	Noise	Noise	1: Not at all to 5: Extremely
	Odors	Odors	1: Not at all to 5: Extremely
	Vibration	Vibration	1: Not at all to 5: Extremely
Living conditions		Please evaluate your living area according to the following items:	
	Green space	Green space?	1: Extremely good to 5: Extremely bad
	Street sceneries	Street scenery?	1: Extremely good to 5: Extremely bad
	View	View?	1: Extremely good to 5: Extremely bad

**Table 8 ijerph-22-01296-t008:** Parameter estimates of the structural equation model for noise annoyance.

Parameter	2019 Survey				2020 Survey				2023 Survey			
	Estimate	SE	CR	*p*	Estimate	SE	CR	*p*	Estimate	SE	CR	*p*
Living conditions ← *L*_den_	0.007	0.004	1.626	0.104	−0.014	0.005	−2.983	0.003	0.000	0.004	−0.089	0.929
Sensitivity ← Living conditions	0.429	0.105	4.104	*	0.050	0.035	1.443	0.152	0.200	0.119	1.683	0.092
Health ← *L*_den_	−0.013	0.003	−4.119	*	−0.001	0.001	−0.643	0.520	−0.004	0.003	−1.366	0.172
Length of time at home ← *L*_den_	−0.006	0.005	−1.287	0.198	0.017	0.004	4.601	*	0.002	0.005	0.373	0.709
Length of time at home ← Living conditions	−0.018	0.078	−0.233	0.816	−0.009	0.048	−0.177	0.859	0.053	0.076	0.695	0.487
Health ← Living conditions	0.099	0.037	2.680	0.007	−0.011	0.016	−0.661	0.509	−0.022	0.038	−0.591	0.554
Health ← Sensitivity	0.085	0.025	3.408	*	0.026	0.039	0.667	0.505	0.166	0.030	5.545	*
Health ← Length of time at home	−0.041	0.022	−1.849	0.064	−0.027	0.040	−0.677	0.499	−0.065	0.030	−2.156	0.031
Health ← Annoyance	0.032	0.010	3.232	0.001	0.004	0.007	0.674	0.500	0.046	0.012	3.964	*
Annoyance ← *L*_den_	0.105	0.013	8.009	*	−0.038	0.018	−2.162	0.031	0.147	0.014	10.402	*
Annoyance ← Length of time at home	−0.001	0.140	−0.006	0.995	1.003	0.276	3.637	*	0.508	0.167	3.035	0.002
Annoyance ← Sensitivity	0.711	0.129	5.494	*	0.254	0.433	0.586	0.558	−0.038	0.110	−0.347	0.728
Annoyance ← Living conditions	−0.141	0.216	−0.652	0.515	−0.071	0.226	−0.314	0.753	0.056	0.223	0.250	0.802

* *p* < 0.001; SE, standard error; CR, critical ratio (CR = estimate/SE).

**Table 9 ijerph-22-01296-t009:** Parameter estimates of the structural equation model for insomnia.

Parameter	2019 Survey				2020 Survey				2023 Survey			
	Estimate	SE	CR	*p*	Estimate	SE	CR	*p*	Estimate	SE	CR	*p*
Living conditions ← *L***_night_**	0.007	0.004	1.635	0.102	−0.014	0.005	−2.829	0.005	0.000	0.004	−0.069	0.945
Sensitivity ← Living conditions	0.428	0.100	4.299	*	0.056	0.037	1.513	0.130	0.200	0.119	1.682	0.093
Health ← *L***_night_**	−0.009	0.003	−3.600	0.065	0.000	0.001	−0.318	0.750	0.001	0.003	0.464	0.643
Length of time at home ← *L***_night_**	−0.007	0.005	−1.370	0.171	0.016	0.004	4.494	*	0.002	0.005	0.420	0.674
Length of time at home ← Living conditions	−0.018	0.077	−0.232	0.816	−0.011	0.048	−0.235	0.814	0.053	0.076	0.695	0.487
Health ← Living conditions	0.078	0.038	2.075	0.038	−0.006	0.019	−0.320	0.749	0.031	0.046	0.673	0.501
Health ← Sensitivity	0.110	0.028	3.876	*	0.015	0.046	0.321	0.748	0.167	0.028	6.023	*
Health ← Length of time at home	−0.053	0.024	−2.204	0.027	−0.012	0.038	−0.322	0.748	−0.103	0.036	−2.847	0.004
Health ← Insomnia	0.206	0.052	3.954	*	−0.006	0.018	0.317	0.752	0.341	0.065	5.204	*
Insomnia ← *L***_night_**	−0.003	0.003	−1.178	0.239	0.010	0.003	3.477	*	0.006	0.003	2.007	0.045
Insomnia ← Length of time at home	0.058	0.029	1.990	0.047	0.038	0.045	0.827	0.408	0.135	0.033	4.054	*
Insomnia ← Sensitivity	0.070	0.029	2.456	0.014	0.118	0.073	1.618	0.106	0.112	0.022	5.060	*
Insomnia ← Living conditions	0.135	0.045	2.983	0.003	−0.002	0.037	−0.045	0.964	−0.039	0.044	−0.870	0.384

* *p* < 0.001; SE, standard error; CR, critical ratio (CR = estimate/SE).

## Data Availability

Data are available from the authors upon reasonable request.

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
