# Peer review of "Assessing Annoyance and Sleep Disturbance Related to Changing Aircraft Noise Context: Evidence from Tan Son Nhat Airport"

_ijerph, 2025, doi:10.3390/ijerph22081296_

Round 1

Reviewer 1 Report

Comments and Suggestions for Authors

It should be detailed how the noise assessment tests were performed.
It should be detailed how the acoustic measurements were performed, how and where the microphone was positioned. The number of measurement points and meteorological conditions (wind and direction) should be described.
Highlight how Lden and Lnight were assessed.
A map of the distribution of Lden and Lnight is missing.
The paper should be simplified.

Reviewer 2 Report

Comments and Suggestions for Authors

Should the authors' contributions to the topic be highlighted?
Your results should be better compared with those of other authors.

Row 104 - Adjust caption position
The same for Fig. 2 and Fig. 3.

Explain in more detail how you obtained the values in Fig. 2 and Fig. 3.

When you administered the questionnaires, how were the participants trained?

Were the tests conducted authentically?

How did you calculate Lden and Lnight?

How did you consider traffic noise or other noises?

Why is only airplane noise annoying?

Airports are generally closed at night.  Why is the airport open in this city during the night?

Aren't there laws prohibiting night flights?

How many points did you consider in the noise measurements?
Did you measure wind speed?

What is the effect of rain?
Did you consider conditions that could alter the measurements?
This part needs clarification.

Row 208.
Explain more clearly how you performed the Relationship Between Exposure Levels and Annoyance.

Row 379. Explain more clearly how you obtained the Annoyance model.

Expand the discussion paragraph.

Increase the references, especially highlighting similar studies.

Round 2

Reviewer 1 Report

Comments and Suggestions for Authors

Accept

Reviewer 2 Report

Comments and Suggestions for Authors

Accept